# Reducing CO₂ Emissions and Improving Water Resource Circularity by Optimizing Energy Efficiency in Buildings

Giada Romano, Serena Baiani [ID], Francesco Mancini *[ID] and Fabrizio Tucci

Planning Design Technology of Architecture Department, "Sapienza" University of Rome, 00196 Rome, Italy; giada.romano@uniroma1.it (G.R.); serena.baiani@uniroma1.it (S.B.); fabrizio.tucci@uniroma1.it (F.T.)
* Correspondence: francesco.mancini@uniroma1.it

**Abstract:** Climate neutrality by 2050 is a priority objective and reducing greenhouse gas (GHG) emissions, increasing energy efficiency, and improving the circularity processes of resources are the imperatives of regulatory and economic instruments. Starting from the central themes of the mitigation of the causes of climate change and the interdependence represented by the water–energy nexus, this research focuses, through the application of the principles of the circular and green economy, on deep energy zero-emission renovation through the improvement of circularity processes of water resources in their integration with energetic ones on the optimization of their management within urban districts, to measure their capacity to contribute towards reducing energy consumption and $CO_2$ emissions during water use and distribution in buildings. After defining the key strategies and the replicable intervention solutions for the circularity of water resources, the investigation focuses on the definition of the research and calculation method set up to define, in parallel, the water consumption of an urban district and the energy consumption necessary to satisfy water requirements and $CO_2$ emissions. Starting from the application of the calculation method in an existing urban district in Rome, 10 indicators of quantities have been developed to define water and energy consumption and their related $CO_2$ emissions, focusing on the obtained results to also define some interventions to reduce water and energy consumption and $CO_2$ emissions in territories that suffer a medium-risk impact from contemporary climatic conditions.

**Keywords:** energy efficiency; climate change mitigation; water–energy nexus; resource circularity; water use and distribution in buildings; reduction in $CO_2$ emissions; optimization of water use; optimization of energy use



## 1. Introduction

Among the various environmental challenges that cities must face, there is the increase in water stress which is limited not only to drought: the excessive extraction of water, especially for urban development, is among the main threats to water resources in the European Union. Since 2012, the European Commission has implemented regulatory instruments, updated according to the water framework directive and aimed at safeguarding water resources, given that the reuse of water has a lower environmental impact, with an important contribution to reducing pollutants and also has potentially lower costs and lower energy consumption when compared to alternative water supplies. The updating of national energy and climate plans since the Green Deal by the member states, which is expected by the end of 2023, must necessarily take into account the new climate objectives because only through the intelligent integration of RES, energy efficiency, and other sustainable solutions in all sectors, will it be possible to contribute to the achievement of the decarbonization objective at the lowest possible cost.

*The Role of Water Resource Circularity in Achieving the Decarbonization Goal*

Before the 20th century, global demand for freshwater was small compared to the natural flows of hydrological cycles [1]. With population growth, industrialization, and

the expansion of irrigation in agriculture, the demand for all water-related needs and services has dramatically increased, jeopardizing the ecosystems that support the water cycle. As demand continues to grow, clean water supply capacities are declining due to increasing pollution of freshwater ecosystems, aquifer water usage, and depletion of aquifer resources [2]. Water also features prominently in the SDGs and plays a central role in various system transitions needed for climate-resilient development; moreover, the achievement of SDG11 (sustainable cities and communities) will necessarily require a reduction in the impacts of water-related disasters [3,4].

Currently, approximately 8 billion people in the world experience severe water scarcity for at least part of the year due to both climatic and human factors. It is, therefore, not surprising that a large share of adaptation interventions (about 60%) are shaped in response to water-related risks [3]. Climate change is affecting water availability, quality, and quantity for basic human needs, threatening the effective enjoyment of the human right to water and sanitation of potentially billions of people [5]. Extreme water-related phenomena exacerbated by climate change are constantly increasing water and sanitation infrastructure risks, such as damaged sanitation systems or flooding of sewage pumping stations [6]. Nowadays, water and wastewater infrastructure in Europe is ageing and requires major investments to prevent even more costly and potentially dangerous system failures and to ensure its resilience in the face of extreme weather conditions [7]. The main policies and strategies of the European Union on climate and energy promote the integration of adaptation to climate change with the safeguarding and protection of natural resources in an increasingly circular perspective: the European Parliament recently approved the recast of the Drinking Water Directive (DWD) [8] which outlines several possible approaches for its implementation in the member states of the European Union, the role played by regulators and water service managers in ensuring the effective adoption of European legislation in the water sector [9], and the need to focus the attention of local agendas on the integrated water system to keep pace with global changes [10].

The European Commission has also launched an evaluation process to review the existing legislation on urban wastewater [11]. According to the new circular economy paradigm, a better and holistic management of wastewater treatment plants must make the best use of resources (e.g., nutrients) and the energy embodied in water [12].

Every phase of the activities of a company that manages water services in urban areas produces carbon emissions: achieving net-zero water and net-zero emission objectives means inevitably including partners that involve the entire supply chain [13]. It is necessary to help cities increase their water resilience [14,15] to build the capacity of urban water systems to resist, adapt, and transform in the face of new climate challenges. For this reason, it is necessary to consider "unconventional" water resources in future planning because, without water-saving technologies, including water reuse, the European Union has estimated a 16% increase in water withdrawal in Europe by 2030 [11,12]. Water reuse is a reliable alternative to conventional water resources for different uses. Although, desalination may also be one of the useful alternatives for increasing the supply of fresh water, this process is generally energy intensive and can contribute to greenhouse gas (GHG) emissions if its energy source is not RES. Despite this, solar desalination is developing rapidly and significantly reduces the carbon footprint of conventional desalination plants [16]. However, although they reduce the carbon footprint, desalination plants release salinity and contaminant levels that are difficult to dispose of and which endanger the health of the earth's surface [17].

Water use based on a linear "take–use–consume–dispose" model, as it has happened historically and up until now, causes a decrease in water quality until it becomes unsuitable for further use both by humans and ecosystems [18]. By the middle of this century, the circular economy has the potential to reduce water consumption from primary resources by 53% [19]. There is growing attention to the treatment of urban wastewater that breaks this linear approach by aiming to clean the water, returning it to the environment for reuse, and allowing the recycling of sewage sludge or the recovery of the resources contained

therein. But focusing only on the treatment process does not lead to circularity, as the principles of minimizing water use and preventing urban wastewater pollution must be applied upstream [20].

Reducing GHG emissions can reduce pollution prevention in urban wastewater treatment plants while improving micro-pollutant removal: "pollutant removal" supports circularity through a wide range of technologies for water reuse and the recovery of energy and resources [21]. Returning water to the environment and recycling it for potable and non-potable use helps regenerate nature: in this direction, it could be possible to achieve circularity in wastewater treatment with both centralized and decentralized approaches. Both systems start with domestic wastewater, but while decentralized treatment conserves resources for local reuse and recycling, the scope of treatment and recovery at major plants can be more intensive [22].

Water cycle circularity, the increase in efficiency in its use, and the reduction in unnecessary water consumption and water losses translate into lower energy consumption and, therefore, lower GHG [23].

The energy sector is a large water user both globally and in Europe. Water consumption by the energy system can compete directly with other water users, such as agriculture and industry. On the other hand, parallel to the energy sector, the water sector is one of the main users of energy, representing about 4% of global electricity consumption [24], and it is precisely both energy optimization in use and the transition to renewable energy sources (RES) which drive utility innovation [25] as part of the race towards zero $CO_2$ emissions. While energy may not be the first thing that comes to mind when it comes to water, the need to pump, pipe, and treat water means that water and energy are always intrinsically intertwined.

The "water–energy nexus" is a term used to describe the interdependence between the two key systems for economic and social development [26].

The interdependence represented by the water–energy nexus (Figure 1) is expected to intensify due to the impacts of climate change, changing consumption and population growth [3–27]. It is also expected that the decline in thermal energy, as predicted in most decarbonization scenarios, leads to reduced water withdrawals; otherwise, higher water consumption by the energy sector in most advanced economies [28] and vice versa, as well as increasing water stress, may limit energy production in some parts of Europe [29,30].

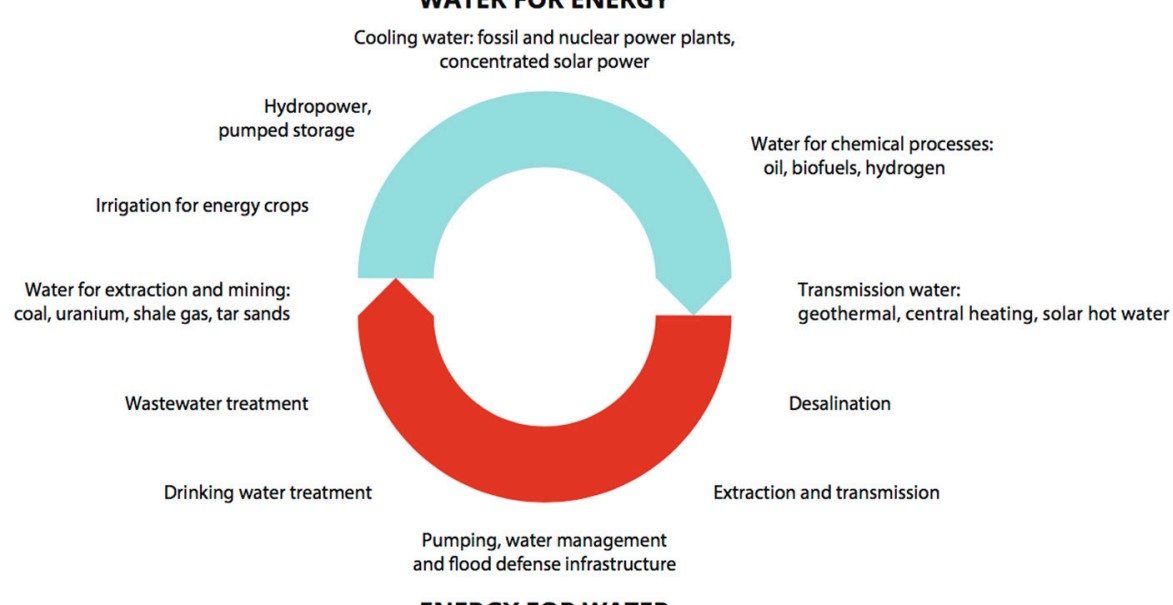

**Figure 1.** Water–energy nexus: the interdependent cycle that links water and energy [26].

Some low-carbon energy technologies have the potential to contribute positively to the nexus, while others can lead to competition or conflict [31]: the replacement of thermal power generation with solar PV and wind energy reduces the use of freshwater per unit of generated energy. At the same time, water use for energy crops and concentrated solar power generation (CSP) can be even higher than that of fossil fuel power plants [32]. Moreover, technologies such as wastewater treatment with energy recovery or biogas production can generate more energy than they use [24–33] and thus lead to positive input.

Energy is essential for the water supply of a wide range of uses which are crucial to the health and well-being of the population, including the supply of safe and clean drinking water, water desalination, wastewater treatment, and pumping and distribution of water for all purposes [22].

The World Environmental and Water Resources Congress study estimated that 7% of world energy consumption in the 2000s came from pumping and water treatment systems for civil use [34]. Reducing energy consumption is essential to reduce $CO_2$ emissions, and utilities will inevitably have to use clean energy sources to resort to decarbonizing energy consumption and achieve the zero-emission goal. Water and wastewater utilities can no longer delay; they have many options to generate clean energy and make their cities' energy mix more ecological and greener. Furthermore, generating energy helps to reduce operational energy costs and grid dependency [35].

Several papers present new perspectives and interpretations of the water–energy nexus, such as exploring the coupling of renewable generation sources with potable water distribution and wastewater harvesting resources capable of storing and releasing energy in response to temporal changes in domestic heating demand [36].

Smart hybrid grey and green infrastructure will be key to achieving climate-neutral and smart urban districts and cities to ensure water supply for essential urban functions and other services, such as energy. Digital solutions are needed to manage multiple water qualities (groundwater, surface, rain, black, and recycled grey) using incentive systems, both through management systems with user involvement and by teaching responsible behaviors in the water smart cities of the future, in combination with circular innovation in the water network [37].

Investments in better water management and new water infrastructure that are more oriented towards the circular economy and green–blue solutions can lead to operational savings: these create value with short payback periods. Cities would also derive multiple environmental benefits from using water to enhance natural capital in all urban spaces. Ponds, streams, and lakes are nature-based solutions that also increase biodiversity; they are more desirable places to live and work next to, improving citizens' quality of life and mental well-being [18]. Nature-based solutions are optimal from two points of view: they provide the water supply necessary for human activities, and they guarantee the functioning of the water cycle and the ecosystem services related to it. However, the lack of a regulatory framework aimed at green–blue infrastructure is one of the main obstacles to accessing public funding. A survey conducted by the OECD shows that although 23 countries out of 27 mention nature-based solutions in their national water management strategy, only 8 demonstrate a direct link with a concrete implementation policy. Italy is among the four countries where nature-based solutions are not mentioned [38].

In parallel with the strategies aimed at the district scale for water infrastructure, which also plays a fundamental role, and the need to move increasingly towards the creation of nature-based solutions and green–blue infrastructure aimed at protecting and recovering natural capital and water resources, there are strategies aimed at the redevelopment of existing buildings with an approach that is ever closer to net-zero water.

After defining the framework for analysis, the first part of this paper circumscribes the investigation's focus through the definition of the strategies, research methodology, and the calculation method set up to define water consumption, the energy consumption from electricity necessary to satisfy water requirements, and the $CO_2$ emissions of the urban district.

Starting from the determination of the calculation method to be applied, 10 indicators of functional quantities have been developed to define, in detail, consumption and the related $CO_2$ emissions into the atmosphere. These indicators have been organized within a calculation tool to estimate the improvements that can be achieved as a function of the possible interventions in an urban district, strictly regarding the optimization of final energy use in buildings thanks to the improvement in water resource circularity strategies and the reduction in water and energy consumption.

## 2. Strategies and Research Methodology

Considering the issues just mentioned, three macro-strategies of intervention have been outlined:

- Water saving and improvement of the water resource circularity process through sustainable management and the "responsible behavior" of users;
- Optimization of the energy efficiency of systems serving the integrated water cycle in buildings and urban districts;
- Reduction in $CO_2$ emissions linked to the integrated water cycle through technological and engineering solutions in buildings and urban districts.

The macro-strategies which lead to the identification of further sub-strategies with the related technological and plant solutions are discussed below.

### 2.1. Water Saving and Improvement of the Water Resource Circularity Process through Sustainable Management and the "Responsible Behavior" of Users

The first of the three macro-strategies of intervention focuses its attention on the two causes of water resource dispersion and waste during urban use and distribution phase: the first strategy concerns the uncontrolled consumption of final users, while the second one consists of the need to limit drinking water consumption to the categories that need it (for personal bathroom and hygiene use; kitchen and dishwasher), which, according to a study on domestic water consumption, account for 47% of consumption [39].

This means that for most of domestic consumption, non-potable water coming from other uses and from the storage, purification, and recovery of wastewater (gray and rainwater) is sufficient. Hence, it is important to combine solution technologies and systems aimed at saving water with those interventions aimed at the collection and circular use of water resources in urban areas which, from the collection of rainwater, also arrives through sustainable urban drainage systems [40,41]. These solutions have the purpose of rebalancing the hydrological cycle, storing, and reusing the wastewater that is sufficient to cover the remaining 53% of domestic water requirements (water supply for air conditioning systems and toilet discharge, irrigation use of crops and green spaces, and cleaning of outdoor spaces). These solutions also aim to reduce pollutants in water and allow cities to behave as "sponge cities" and exploit the ecosystem services that derive from nature-based solutions.

The design and installation of technological solutions aimed at both water saving and circular management of water resources must necessarily be accompanied and supported by expedients, habits, and good practices, which, like the technological solutions, are equally aimed at water saving, circular management, and water footprint reduction.

### 2.2. Optimization of the Energy Efficiency of Systems Serving the Integrated Water Cycle in Buildings

Once the first macro-strategy has been outlined, with its related intervention solutions, habits, good practices, and attitudes aimed at sustainable management and the improvement of the circularity processes of water resources, another technical aspect to consider is linked to the obsolescence of the plant devices that are still in use, as compared to the modern and more efficient ones, and to an efficiency improvement of the integrated urban water cycle to reduce the water footprint and $CO_2$ emissions, thus improving efficiency both at the local scale of the building and at the urban scale of an urban district.

Digitization and energy efficiency are the basis of the responses necessary to face the emerging challenges and objectives of the water industry, which include the following: a

guarantee of the quality of the water resource, growing water demand, the consequences of climate change, end-user life quality improvement, infrastructure degradation and water asset maintenance, leak detection, efficiency enhancement, tariff sustainability, and compliance with regulations and adjustment.

Cities need to consider how to match water quantity and quality to intended use, looking at the entire water cycle from a prospective view, where all water is useful both to water and resource recovery and energy efficiency of the systems serving the integrated water cycle.

The efficiency of water services involves different and synergistic actions that can bring advantages on three interconnected fronts: water saving, energy saving, and energy production. The more efficient energy recovery measures implemented within the water cycle, in some cases, can allow compensation for the water service energy needs or, in any case, bring this sector closer to zero energy consumption. A further level of optimization of the water system concerns the possibility that it can be managed as a flexible element of the electricity system [42], allowing the absorption of electricity generation peaks from non-programmable RES [43].

Energy efficiency in the various phases of the integrated water cycle also requires an appropriate choice of RES type to be inserted according to the environmental parameters linked to water resources and to the geographical position of the various urban districts. Renewable energy plays a fundamental and ever-growing role in the European and global energy system, and energy efficiency maximization derives from the type of RES that is mainly widespread, according to the geo-morphological characteristics and site's climate.

### 2.3. Reduction in $CO_2$ Emissions Linked to the Integrated Water Cycle through Technological and Engineering Solutions in Buildings

Analyzing the regulatory instrument introduced by ARERA in 2017 and the use of the UNI 14064 standard for calculating the reductions in climate-changing gas emissions into the atmosphere, in terms of reduction in $CO_2$ emissions in the urban district, the indirect emissions to consider (as emissions attributable to operational, managerial, or operational choices of the organization) are the emissions for energy consumption. These emissions generally represent one of the main contributions to the total emissions of the integrated water system; therefore, the primary decarbonization interventions must be aimed primarily at reducing energy consumption [44].

The technical and scientific literature reports various tools and software developed to estimate the carbon footprint of integrated water services. Stationary calculation and dynamic simulation tools have been implemented for determining the carbon footprint of a single plant or purification service and for calculating direct emissions; however, the same models fail to consider the entirety of the climate-changing emissions from the plants because they do not consider the aerated treatment units and consequently provide a complete but not exhaustive picture [44,45].

Only the $CO_2$ emissions of fossil origin are generally considered; for the biogenic component, although also in the water sector, a significant share deriving from microbial respiration during biological processes is usually not quantified.

The same IPCC guidelines [46] do not consider $CO_2$ emissions of biogenic origin since it is believed that they have no impact on the increase in $CO_2$ in the atmosphere for assessing the impacts of human activity on atmospheric variations.

The definition of the actions to optimize processes and treatments aimed at improving the circular process and the operating parameters make a reduction in $CO_2$ emissions possible. The reduction could be possible through the implementation of circularity processes with passive solutions and predominantly installations or with passive aeration technology in terms of the filtration and purification sectors, while, regarding treatment, the purification and distribution through the control of energy consumption in pumping systems and the monitoring of distribution grids aimed at reduction in water losses. It could also possible with the installation of RES to be produced and consumed on-site,

in particular solar and hydro-electric ones aimed at reducing emissions due to electricity consumption through the replacement or revamping of existing systems, the energy upgrading of buildings, and the application of water management systems that can manage the demand for the required resource and load capacity with a view toward forecasting that permits a reduction in energy consumption.

*2.4. Research Method: Calculation of Water Consumption, Energy Consumption Linked to Water Needs, and CO$_2$ Emissions*

The calculation methodology has been developed considering the ongoing experiments of circular and climate-neutral cities and analyzing the international and national regulatory frameworks that push towards the definition and application of improvement parameters in the use of water in buildings through the implementation of collection–recovery–reuse of wastewater and the defined macro-strategies aimed at maximizing the role of water resources in achieving the objectives of mitigating the causes of climate change and urban district decarbonization.

Starting from water consumption calculation and energy consumption linked to water needs would also allow for an assessment of CO$_2$ emissions considering the overall picture of the integrated water cycle in the drinkable water distribution and use phases inside the buildings. The first step toward reducing CO$_2$ emissions is to quantify the carbon footprint. This step has only been possible after analyzing the methods of calculating and quantifying the water needs of buildings in the sustainability protocols ITACA, LEED, and BREEAM, allowing an evaluation of which of these could be the most suitable for use as a reference to calculate the indicators.

2.4.1. Calculation Method Adopted for the Water Requirements of Buildings

The calculation method has been reworked and implemented for this discussion. Starting from the water consumption calculation methods of the ITACA protocol [47,48] and from the in-depth study of manuals aimed at the design and construction of hydraulic works [49] to analyze water and energy needs of the urban district, they have been divided into six indicators of the quantity of water consumption:

1.　Amount of drinkable water for indoor use in residential buildings;
2.　Amount of drinkable water for indoor use in non-residential buildings;
3.　Amount of wastewater sent to the district sewage system;
4.　Amount of rainwater captured and stored;
5.　Amount of water needed to irrigate green areas;
6.　Quantity of water losses from the water grid.

It was possible to obtain the seventh indicator of the quantity of overall water consumption and the eighth indicator of the amount that allows us to analyze the energy consumption necessary to cover the global water requirement:

7.　Overall water requirement in buildings;
8.　Amount of electricity to cover the water needs of the buildings.

Once the overall water requirement and the correlated energy consumption have been calculated, it is possible to analyze water demand management regarding CO$_2$ emissions in the aqueduct segment of the integrated water cycle during the distribution and use phases at the district level [50].

Through the analysis of the quantity of CO$_2$ emitted, it was possible to obtain the indicators of the amount of CO$_2$ emissions:

9.　Amount of CO$_2$ emissions related to water needs of the urban district;
10.　Amount of CO$_2$ emissions related to the production of electricity to cover the water needs of the urban districts.

2.4.2. Calculation of the Overall Water Requirements in Buildings

Having the first five indicators, the global water requirement of the urban district can be calculated by adding up the individual contributions, according to the following formula:

$$F_{wat,tot} = F_{indoor,\ res} + F_{indoor,\ n\ res} + F_{non-dr} + \max[F_{wat,irr}; F_{wat,irr\_eff}]\ , \tag{1}$$

with

$F_{wat,tot}$ = overall water requirement [m$^3$/y];
$F_{indoor,res}$ = drinkable water for indoor use in residential buildings [m$^3$/y];
$F_{indoor,n\ res}$ = drinkable water for indoor use in non-residential buildings [m$^3$/y];
$F_{non-dr}$ = non-drinkable water [m$^3$/y];
$F_{wat,irr}$ = water needed to irrigate green areas [m$^3$/y];
$F_{wat,irr\_eff}$ = effective water needed to irrigate green areas [m$^3$/y].

2.4.3. Calculation of the Amount of Electricity to Cover the Water Needs of Buildings

The eighth indicator is the amount of electricity to cover the water needs of buildings. This indicator, although in a decidedly small percentage when compared to the quantity of electricity necessary to satisfy the uses of HVAC systems, DHW production, and lighting, nevertheless has a weight that must be considered, especially in light of the fact that the use of energy from fossil fuels in water infrastructure throughout the integrated cycle is responsible for 52% of the sector's emissions [35].

To carry out this calculation, however, there is no univocal datum; for this reason, it has been necessary to determine a further variable parameter both on an annual basis and according to the geographical location of the urban district being considered. A conversion index (CI = k$_4$) of water consumption into electricity consumption is indispensable; it has been calculated according to the following formula:

$$CI = k_4 = \frac{Q_{el\_IWS}}{F_{wat,tot\_nat}}, \tag{2}$$

with

$k_4$ = national conversion index [kWh/m$^3$];
$Q_{el\_IWS}$ = national annual electricity consumption of the integrated water system [kWh/y];
$F_{wat,tot\_nat}$ = national annual water consumption [m$^3$/y].

Given that the variability of water and electricity consumption for water use and distribution data depends on the nation in which they are analyzed, a table was created with the data that made it possible to calculate the conversion index for 12 European countries that have been reported, starting from the respective consumption recorded at the national level for the year 2020 [51–53].

Table 1 presents a picture of the situation of electricity and water consumption in 2020 for the 12 countries considered as examples. Starting from the identification of the conversion index, we proceeded with the calculation of the amount of electricity needed to satisfy the water needs in urban districts, according to the following formula:

$$Q_{el\_wat} = k_4 \cdot F_{wat,tot}, \tag{3}$$

with

$k_4$ = national conversion index [kWh/m$^3$];
$Q_{el\_wat}$ = electricity needed to satisfy the water requirements [kWh/y];
$F_{wat,tot}$ = overall water requirement [m$^3$/y].

This indicator is also functional for calculating the amount of electricity needed to meet the water needs of the urban district per capita per inhabitant per day, which is particularly relevant for a comparison with global energy production electricity for other uses commonly considered at the district level (HVAC, DHW production, and lighting) to determine, depending on the result obtained, the percentage of incidence for the total.

**Table 1.** National electricity consumption [million kWh/y], water consumption [million m³/y], and conversion index $k_4$ [kWh/m³] for 12 European countries (data from [51–53]).

| Country | Electricity Consumption [Million kWh/y] | Water Consumption [Million m³/y] | $k_4$ [kWh/m³] |
|---|---|---|---|
| Austria | 190 | 577 | 0.329 |
| Belgium | 658 | 738 | 0.891 |
| Denmark | 144 | 360 | 0.400 |
| Finland | 257 | 322 | 0.797 |
| Ireland | 178 | 329 | 0.541 |
| Italy | 7500 | 26,000 | 0.288 |
| Netherlands | 456 | 1140 | 0.400 |
| Portugal | 531 | 572 | 0.929 |
| Spain | 1218 | 3107 | 0.392 |
| Sweden | 473 | 675 | 0.700 |
| Switzerland | 351 | 761 | 0.461 |
| Hungary | 256 | 454 | 0.563 |

### 2.4.4. Calculation Method of the Amount of $CO_2$ Emitted by the Integrated Water Cycle during the Use and Distribution Phase

Starting from the eight indicators that define, individually and as a whole, the global water requirement and the number of water losses in the phases of distribution and use in the urban districts and from literature studies especially from the methodological approach reported in [50] which made it possible to quantify the $CO_2$ emitted in the aqueduct segment of the integrated water cycle during the distribution and use phases linked to water consumption and the emissions related to electricity consumption, the calculation method has been set.

As in the case of calculating electricity consumption, in this case, it is necessary to have a given quantitative index which allows obtaining the specific $CO_2$ related to water consumption and further quantitative index data that allow obtaining the $CO_2$ emissions related to the production of electricity necessary to satisfy the water needs of the urban district. These indices vary substantially because the integrated water service, in its complexity and fragmentation, represents a service whose environmental performance generally depends on the topography of the area, the sources of water supply, the population density, the resident population, and the distance of the buildings occupied by end users in terms of inhabited centers and by the centralization of treatment plants as well as further characteristic factors of each portion of the territory. Consequently, the necessary quantitative indices once again constitute a variable element that could be checked in the sustainability reports or in the information made available by the managers of the reference water service.

Finally, the variability of the unitary quantitative index considered depends on the single phase of the water service, the type of emission considered in the calculation based on the application of the ISO 14064 standard (direct, indirect, biogenic, or non-biogenic) and the ISO 14067 standard, and the reference calculation method used ("cradle-to-gate" up to the production phase, the most common approach in the water service or "cradle-to-grave", i.e., coming to consider emissions up to the end-of-life stage).

### 2.4.5. Calculation of $CO_2$ Emissions Related to the Production of Electricity Necessary to Cover the Water Needs

For the calculation of the $CO_2$ emissions linked to the production of electricity necessary to satisfy water needs, steps have been taken to identify an additional quantitative index, again starting from the information provided by the water service operators, calculated in compliance with the provisions of the ISO 14064 standard as indirect emissions deriving from the consumption of electricity for the production, use, and monitoring of systems and devices serving the integrated water cycle in the use and distribution phase. This index varies according to each urban water service operator's consumption for each compartment under its jurisdiction, and it can change for every urban district.

Once the quantitative index (QI = $k_7$) was identified for the calculation of $CO_2$ emissions linked to the production of electricity necessary to satisfy the water needs of the urban district, we proceeded by applying the following formula:

$$CO_{2\_el\_wat} = k_7 \cdot Q_{el\_wat},\tag{4}$$

with

$CO_{2\_en,el\_idr}$ = $CO_2$ emissions related to the production of electricity necessary to cover water needs [kgCO$_{2eq}$*m$^3$/y];

$k_7$ = quantitative index of the $CO_2$ released into the atmosphere during the production of electricity related to the use and distribution phases in urban districts [kgCO$_{2eq}$/kWh];

$Q_{en,el\_idr}$ = electricity needed to satisfy water requirements [kWh/y].

*2.5. Optimization of the Energy Use Related to Water Consumption in Buildings: Three Renovation Scenarios*

The calculation method allows for energy use optimization related to water consumption in buildings and urban districts with specific solutions that respond to the macro-strategies.

Based on the achievement of a reduction in consumption and emissions, three different scenarios have been developed: light energy zero-emission renovation (broken down into possible interventions by end users and necessary interventions by water utilities), medium energy zero-emission renovation, and deep energy zero-emission renovation.

The three scenarios differ in the definition of strategies and simulation of the effects of solutions corresponding to the light, medium, and deep levels aimed at reducing water, energy, and $CO_2$ emissions.

In the following three scenarios, average costs per square meter for natural water retention interventions have been referenced for cost analysis related to sustainable urban drainage techniques. Finally, for items missing in each price list consulted, added price analysis was used considering quotes from supply and production companies for the technological and plant engineering devices envisaged by the project.

Strategies and solutions are better explained and more specified in the next three paragraphs.

2.5.1. Scenario 1: Light Energy Zero-Emission Renovation

The first scenario of light energy zero-emission renovation has two levels of possible interventions: for the users or necessary for the water service operators that are split into scenario 1A and scenario 1B, which are to be combined again in the returning data phase obtained from the proposed interventions.

The intervention strategies defined for scenario 1A of light energy zero-emission renovation are articulated in the possible design solutions that involve a minor impact at the scale of the building, both from the point of view of the redevelopment works, which in this case involves interventions of a non-structural and economic nature.

This scenario evaluates measurable technological and plant interventions that provide for a reduction and rate control of the flow and temperature of water in buildings, made possible by users and water service managers.

The intervention strategies defined for scenario 1B of light energy zero-emission renovation are the design solutions required by the pertinent water service managers. Although they certainly have a more notable impact than the solutions proposed in scenario 1A, they are in any case attributable to intervention strategies that produce a minimal environmental impact on the territory. The reference macro-strategies, in this case, are both the improvement of the circularity of water resources through sustainable management by the water service managers and the maximization of energy efficiency of the active systems serving the integrated water cycle of the urban district with a consequent reduction in $CO_2$ emissions linked to the integrated water cycle with technological and plant solutions at the scale of the urban district.

Scenario 1 has two redevelopment scenarios: scenario 1A and 1B, with the specific purpose of bringing out the two aspects related to the non-structural design interventions made possible by the inhabitants and end users and those with the low environmental impact required by the local water service operators. Scenario 1A shows how, through the installation of technological and plant engineering devices involving flow reduction and control of water flow rate and temperature, it is possible, without structural interventions and with an average cost varying between EUR 1500 and EUR 2500 per user, to achieve a percentage reduction in both water consumption and related $CO_2$ emissions of between 10% and 20%.

This percentage, although not so high, still leads to a tangible reduction in per capita water demand per day to 200 L, a value that, although high, is lower than that estimated in the business-as-usual situation, even more so when evaluated over an annual period. However, with scenario 1A, there is no resource circularity improvement but only a reduction in consumption. On the other hand, scenario 1B, in light of the actual targets and those achieved so far between 2019 and 2021 in terms of reducing water leakage and the resulting $CO_2$ emissions, declared in the sustainability report by the local water utility, shows how, even if only by acting on the water supply with a view toward districting and limiting leakage during use and distribution, can achieve a reduction in water consumption and related $CO_2$ emissions of more than 50%. This percentage underscores the greater weight of over-districted water infrastructure management as compared to that of utilities, even if only considering the use and distribution phase of the aqueduct segment.

### 2.5.2. Scenario 2: Medium Energy Zero-Emission Renovation

The second scenario of medium energy zero-emission renovation adds to the strategies and intervention solutions outlined in the renovation interventions of scenario 1 aimed at improving the circularity processes of water resources.

The intervention strategies for scenario 2 are split into design solutions, sometimes structural, which are possible at the scale of buildings and urban districts, with an average expenditure of about EUR 1500 per square meter to be added to the cost for the first scenario and for the purchase and installation of tanks and plant engineering to capture and store water sources.

Thanks to the combination of the possible interventions at the scale of buildings and of urban districts in scenario 1 and scenario 2, we arrive at the definition of an almost complete process of circularity of the water resource (which includes the re-entry into circulation and the reuse of both the components of rainwater and gray water), the reduction in water consumption, the containment of network losses, and the proportional increase in the levels of electrification of active devices aimed at the circularity of the water resource and the smart monitoring of consumption and water losses.

These solutions also indirectly lead to a reduction in $CO_2$ emissions thanks to the optimization of processes and treatments aimed at improving circularity and operating parameters but do not intervene on the sources of energy production necessary to guarantee their functioning.

With the upgrading interventions provided in medium energy zero-emission renovation scenario 2, considerable results have been achieved in terms of reducing overall and per capita water demand, leakage (already achieved through the interventions provided in scenario 1B), and energy consumption and related $CO_2$ emissions. Specifically, drinking water consumption per capita per day can be reduced to 40 L, which is a percentage reduction of 80% as compared to the consumption in the previous scenario 1 and 83% as compared to the status quo. Such a high percentage reduction from the current situation would never have been achievable if not for the implementation of technological and plant solutions aimed at the accumulation, storage, and reuse of meteoric wastewater components, wastewater from sinks, showers, and tubs, and condensate from air conditioning systems. These are all valuable components of the water resource that in the previous scenario had been fed into the district sewage system and instead, in this framework of design interventions, are

recirculated to cover the portion of indoor needs that do not require the use of potable water as well as for irrigation use.

The circular use of the water resource through the reuse of wastewater contributes both to a reduction in the volumes extracted from aquifer sources and to a reduction in the irresponsible and unmotivated spillage of water of possible "second use" into urban sewage systems, creating a further overload. Also complicit in this scenario of a reduction in drinking water consumption and withdrawals, are district-level interventions aimed at increasing permeable surfaces in pedestrian and parking areas in districts, as well as on building roofs and sustainable urban drainage, which together contribute to additional storm water storage for irrigation of public green areas, purification, and natural infiltration into the ground, thus limiting surface runoff levels and associated episodes of flooding during extreme weather/climate events.

2.5.3. Scenario 3: Deep Energy Zero-Emission Renovation

The last scenario, scenario 3 of deep energy zero-emission renovation, was used to set up the two previous scenarios and encompasses the strategies and technological and plant solutions aimed at improving the circularity process of the water resource and maximizing the energy efficiency of systems serving the integrated water cycle of urban districts and the reductions in $CO_2$ emissions into the atmosphere by urban districts.

The intervention strategies defined for scenario 3 are divided into possible design solutions by the end users and can be implemented both at the building and at the urban district scale, with an average cost that comes from the addition of the previous scenarios and the installation of RES plants.

The plant intervention that can be implemented to reduce, up to zeroing, $CO_2$ emissions during use and distribution is attributable to power from RES to be produced and consumed on-site. This type of intervention can vary substantially according to both the criticality of the zone where the urban district is located, from the point of view of the impact of climate change on water resources, and the environmental analysis of shading, sunshine, and ventilation carried out for the assessment of the most efficient RES.

In the specific case of a study of national and international reports on the state of renewable energy sources in Italy [54,55], PV solar energy is the best choice, from the point of view of the efficiency of the system being considered and given the practicality and economy of installation costs.

The creation of a PV solar system, integrated into existing buildings and, only where necessary, placed on the roofs, involves the production of electricity to cover the water needs of the urban district and the reduction in related $CO_2$ emissions that can reach zero, considering the emissions during use and distribution of the integrated water service.

Scenario 3 of deep energy zero-emission renovation is the most comprehensive. In this scenario, the energy consumption required to meet the water needs of districts is acted upon, and it is through this type of intervention that the core goal of this research, i.e., the decarbonization of the building stock and urban districts, is met for the portion that relates to water-related energy consumption.

**3. Results and Discussion**

The calculation method, the strategies, and solutions articulated in the three intervention scenarios have been applied to one of the Roman affordable and social housing districts: the INCIS-Decima District. This urban district has an area of about 22 hectares located southwest of the EUR District and is characterized by a rationalist architecture on pilotis buildings, built between 1957 and 1965.

The urban conformation of the district presents five building types: serial comb buildings arranged in the southwest area, margin buildings to the southeast, L-shaped and C-shaped margin buildings placed to the northwest, and fifth buildings composing the central spine. The volumetric composition of the buildings is by "islands", where the buildings are shaped in concave and convex forms.

The INCIS-Decima District, currently located in the ninth Municipality of the Metropolitan City of Rome Capital (formerly the 12th Municipality of Rome), has 808 apartments with 5724 rooms in buildings.

A first analysis of the demographic and spatial and volumetric composition of the neighborhood (Table 2), carried out with the population and housing census information conducted by the Statistics Operational Unit of the Municipality of Rome, has made it possible to identify the prevalence of residential use over other uses (office, commercial, receptive, educational, and restaurant use), which do not appear to be concentrated in individual buildings but spread out spatially.

**Table 2.** Reconstruction of the spatial configuration of the neighborhood and the number of inhabitants and end users by intended use.

| Intended Use | | Buildings | Inhabitants | Surface [m$^2$] | Volume [m$^3$] |
|---|---|---|---|---|---|
| Residential | | 30 | 3680 | 54,528 | 163,584 |
| Non-residential | Office | 1 | 6530 | 6530 | 19,590 |
| | Commercial | 2 | 1053 | 7376 | 22,128 |
| | Receptive | 1 | 276 | 6072 | 18,216 |
| | Educational | 3 | 1368 | 10,940 | 32,820 |
| | Restaurants | 1 | 295 | 1196 | 3588 |
| Total | | 38 | 7326 | 86,642 | 259,926 |

In order to build the simulation model for the calculation, the choice has been to reconstruct the spatial and volumetric composition of the neighborhood from the sum of the areas allocated to the various residential and non-residential uses.

The results obtained with the simulation highlight that is only thanks to the combination of interventions, in line with the green city approach, including reduction and control of water consumption, responsible behavior, sustainable use and conscious management by users, containment of losses from the water grid, improvement of the circularity process of the water resource, electrification of technological devices and systems for smart management and remote monitoring, and modification of the source of electricity production, strictly on-site, which contributed, through a deep energy zero-emission renovation of the existing building stock and of the urban districts, to the achievement of the decarbonization objective with a strong impact on the theme of mitigating the causes of climate change.

Below are the graphs showing the results obtained in three redevelopment scenarios in comparison with each other and with the current situation as regards to water consumption, energy consumption, and $CO_2$ emissions into the atmosphere by the district.

The breakdown of water consumption in the urban district, in the graph and table below (Table 3 and Figure 2), highlights the consumption linked to the water requirement of drinking water for indoor use (which derives from the sum of residential and non-residential uses and which is graphically represented by the first columns of each criterion), consumption linked to the quantity of drinking water poured into the district sewage system (second columns of each scenario), consumption of water for non-potable uses and irrigation use (third columns of each scenario), losses from the water distribution network (fourth columns of each scenario), and global water consumption (fifth columns of each scenario); this consumption, as already analyzed individually for the first two scenarios, is reduced, initially slightly and then substantially, as compared to the current situation, to then remain unchanged in the third scenario.

Following the consumption of water, the table and graph below (Table 4 and Figure 3) represent the distribution of energy consumption in the urban district, divided between residential use (first columns of each scenario), non-residential use (considered together and shown by the second columns of each scenario), and global energy consumption (which derives from the sum of consumption for residential and non-residential uses and which are graphically represented by the third columns of each scenario).

**Table 3.** Table showing the water consumption of the INCIS-Decima District in the light energy zero-emission renovation (scenario 1), medium energy zero-emission renovation (Scenario 2), and deep energy zero-emission renovation (Scenario 3) interventions, in comparison with the current situation.

| Water Consumption [m³/y] | $F_{indoor\_eff}$ | $Eff_{indoor\_eff}$ | $F_{non\ pot} + F_{irr}$ | Losses | $F_{idr\_gl}$ |
|---|---|---|---|---|---|
| Current situation | 198.3 | 148.4 | 239.4 | 173.7 | 607.4 |
| Scenario 1 | 156.2 | 118.7 | 239.4 | 82.0 | 535.5 |
| Scenario 2 | 26.3 | 48.3 | 30 | 35.2 | 105.1 |
| Scenario 3 | 26.3 | 48.3 | 30 | 35.2 | 105.1 |

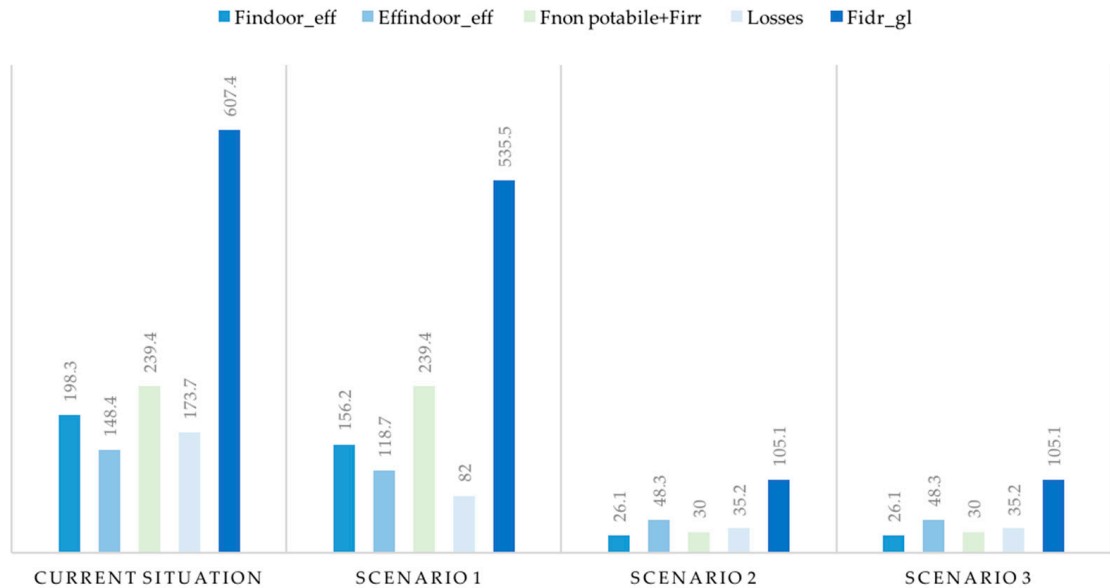

**Figure 2.** Graph showing the water consumption of the INCIS-Decima District in the light energy zero-emission renovation (Scenario 1), medium energy zero-emission renovation (Scenario 2), and deep energy zero-emission renovation (Scenario 3) interventions, in comparison with the current situation.

**Table 4.** Table showing the electricity consumption of the INCIS-Decima District in the light energy zero-emission renovation (scenario 1), medium energy zero-emission renovation (Scenario 2), and deep energy zero-emission renovation (Scenario 3) interventions, in comparison with the current situation.

| Electricity Consumption [kWh/y] | $Q_{idr,res}$ | $Q_{idr,n\ res}$ | $Q_{idr,gl}$ |
|---|---|---|---|
| Current situation | 132,327 | 42,606 | 174,933 |
| Scenario 1 | 117,317 | 36,919 | 154,236 |
| Scenario 2 | 25,015 | 9751 | 34,765 |
| Scenario 3 | 25,015 | 9751 | 34,765 |

Given the strong link that correlates to water consumption, energy consumption follows the same trend or decreases as compared to the current situation, initially slightly and then consistently for the first two renovation scenarios, to remain unchanged in the third scenario.

Although from the point of view of water consumption and losses from the distribution network no additional interventions have been planned, scenario 3 of deep energy zero-emission renovation, is the most comprehensive. This scenario acts upon the energy consumption required to meet the water needs of the urban district.

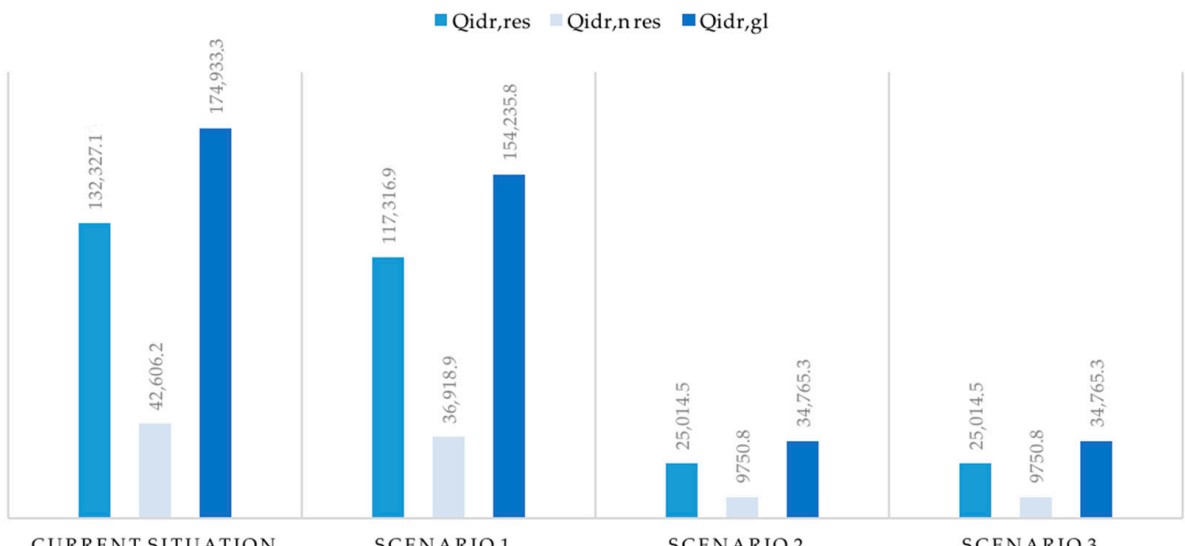

**Figure 3.** Graph showing the electricity consumption of the INCIS-Decima District in the light energy zero-emission renovation (scenario 1), medium energy zero-emission renovation (scenario 2), and deep energy zero-emission renovation (scenario 3) interventions, in comparison with the current situation.

The conversion from non-renewable to RES of the source of electricity production, thanks to the implementation of a PV system integrated into existing buildings (together with the electrification of smart technological solutions and devices aimed at the remote management and monitoring of water resources as well as pumping and distribution systems of re-circulated water, envisaged in the previous scenarios) leads, against the reduction in consumption already achieved in medium energy zero-emission renovation scenario 2 and thanks to the improvement in the water resource circularity process, to a total zeroing of related $CO_2$ emissions and thus to a 100% reduction when compared to the current situation.

The following table and graph (Table 5 and Figure 4) represent the $CO_2$ emissions into the atmosphere linked to the energy consumption needed to cover water consumption, divided residential use (first columns of each scenario), non-residential use (all taken together and represented by the second columns of each scenario), and global $CO_2$ emissions (which result from the sum of consumption for residential and non-residential uses and which are graphically shown by the third columns of each scenario); the $CO_2$ emissions, as already analyzed individually and in part anticipated with regard to the water consumption graph, are reduced for the first two scenarios, initially slightly and then substantially, as compared to the current situation and then reach zero completely in the third scenario of deep energy zero-emission renovation.

**Table 5.** Table showing the $CO_2$ emissions related to the production of electricity necessary to cover the water needs of the INCIS-Decima District in the light energy zero-emission renovation (scenario 1), medium energy zero-emission renovation (scenario 2), and deep energy zero-emission renovation (scenario 3) interventions, in comparison with the current situation.

| $CO_2$ Emissions [kgCO$_{2eq}$*m$^3$/y] | $CO_{2\_idr,res}$ | $CO_{2\_idr,n\ res}$ | $CO_{2\_idr,gl}$ |
|---|---|---|---|
| Current situation | 693.4 | 223.3 | 916.7 |
| Scenario 1 | 614.5 | 193.5 | 808 |
| Scenario 2 | 114 | 44.4 | 158.4 |
| Scenario 3 | 0 | 0 | 0 |

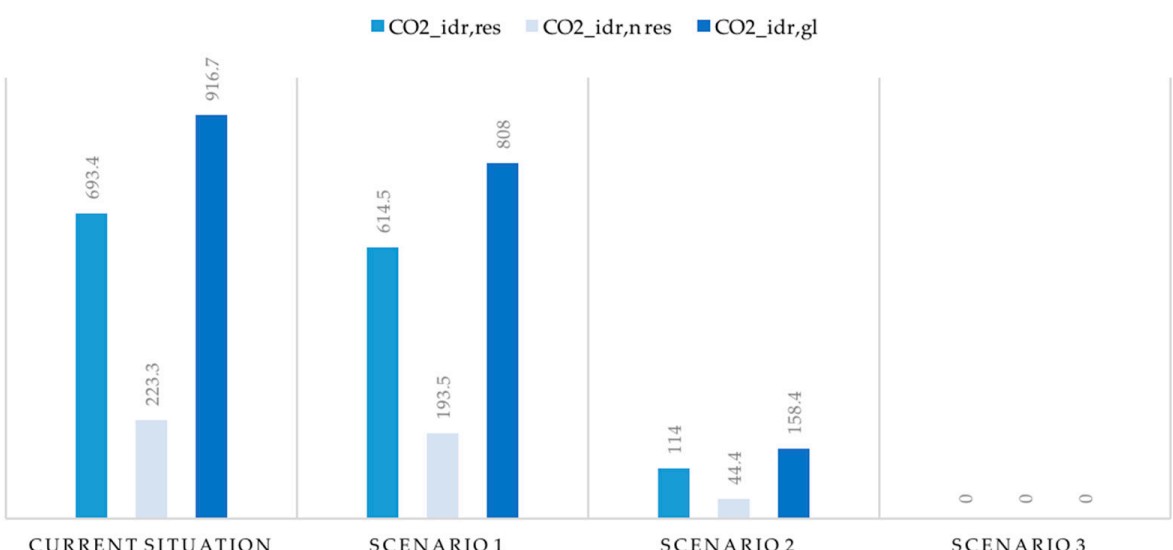

**Figure 4.** Graph showing the $CO_2$ emissions related to the production of electricity necessary to cover the water needs of the INCIS-Decima District in the light energy zero-emission renovation (scenario 1), medium energy zero-emission renovation (scenario 2), and deep energy zero-emission renovation (scenario 3) interventions, in comparison with the current situation.

The results obtained from the simulation of the three scenarios show that the zeroing of $CO_2$ emissions related to water and energy consumption obtained in the third scenario of deep energy zero-emission renovation and the achievement of the decarbonization objective of the existing building stock, thanks to the improvement of the circularity process of the water resource, concerns only a percentage equal to about 3.7% of global energy consumption, as revealed by the comparison of the energy consumption obtained from that defined by the reference benchmarks for each intended use. It is as much a part of consumption and climate-changing emissions when considering the entire district, which often risks being overlooked or underestimated in global consumption and emission analyses.

## 4. Conclusions

Knowing that there are different ways to optimize energy efficiency in buildings, the novelty of the study is that it focuses, through the application of the principles of the circular and green economy, on deep energy zero-emission renovation, through the improvement of the circularity processes of water resources in their integration with energetic ones for the optimization of their management within the urban districts, to measure their capacity to contribute towards the reduction in energy consumption and $CO_2$ emissions during water use and distribution in buildings. Above all, this study has demonstrated that none, or almost none, of the anthropic components as opposed to natural or energy resources, alone, can make a single contribution that is so consistent as to be able to satisfy the prefixed objective of zeroing $CO_2$ emissions by 2050; it is only thanks to a systemic and synergistic combination of blue, green, and gray strategies and the application of passive environmental–technological systems and plant systems, that this aspect, in quantitative terms, can be considered fully satisfied.

Water is a source of life, but at the same time, it is under increasing pressure: water stress and over-extraction from natural sources are among the various environmental challenges facing cities in the short-term. The optimal management of the precious water resource must also become an unavoidable goal considering climate change impacts on the environment, but there is still too little awareness on how to pursue it. Optimal water

resource management must be placed in a broader perspective, addressing all types of users and taking account of water interactions with other resources, such as soil and energy.

Furthermore, operating on water resources requires an essential involvement of site-specific water service management and, in turn, each simulation requires quantitative input indicators to obtain a compliant picture of the actual emissions situation. Indicators are not frequently calculated (or disclosed) by water managers; on the contrary, they are relatively rare to find and, from time to time, they can undergo even considerable variations, depending on the orography of the territory, the water supply and energy production sources typical of the local integrated water service, the population density of the district or urban district considered, and the distance of the users from the water distribution systems. Therefore, it is only by starting from the definition of the unitary indicators that it is possible to identify and calculate the $CO_2$ emissions released into the atmosphere by each urban district; these quantities are necessary to enter the emissions framework achieved, not only in the set of urban district global emissions but also in those relating to the water infrastructure.

However, the technological strategies and solutions, defined in the deep energy zero-emission renovation scenario and designed to eliminate $CO_2$ emissions related to water and energy consumption in quantitative terms, allow a reduction of slightly less than 5%, corresponding to the $CO_2$ emissions not released into the atmosphere as a function of water demand, in the use and distribution phases. This is a small but essential percentage, as compared to the global $CO_2$ emissions of urban districts, if we want to reach the net zero energy district and the net zero carbon or carbon-neutral district paradigms by systematizing green, blue, and grey components and systems responsible for climate-changing gas emissions.

Following up on these considerations, it is evident how far this study has, on the one hand, arrived in terms of defining a calculation tool and a series of macro-strategies and design intervention solutions outlined promptly and whose impacts are quantifiable, but on the other, it leaves a series of open issues related to the difficulty of isolating a fixed unitary parameter of the amount of $CO_2$ emitted into the atmosphere from the water distribution phase alone and from the entire integrated water service, as well as the impossibility of finding solutions to replicate in every urban context without considering the impacts that climate change entails on the environmental parameters most closely related to the water resource. Open issues that not only concern the present study but also reflect an international framework that is constantly evolving as well as increasing attention to water infrastructure, a sector in which, among other things, a real revolution in terms of reporting on environmental, social, and governance impacts is expected by 2025.

**Author Contributions:** Conceptualization, G.R. and S.B.; methodology, F.M. and G.R.; software, F.M. and G.R.; validation, F.M. and S.B.; writing—original draft preparation, G.R., S.B., F.M.; writing—review and editing, G.R.; visualization, G.R.; supervision, F.T., F.M. and S.B.; project administration, F.T. All authors have read and agreed to the published version of the manuscript.

**Funding:** This research received no external funding.

**Institutional Review Board Statement:** Not Applicable.

**Informed Consent Statement:** Not Applicable.

**Data Availability Statement:** Not Applicable.

**Conflicts of Interest:** The authors declare no conflict of interest.

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
