# Peer review of "Reducing CO2 Emissions and Improving Water Resource Circularity by Optimizing Energy Efficiency in Buildings"

_sustainability, doi:10.3390/su151713050_

Round 1

Reviewer 1 Report (Previous Reviewer 2)

revised manuscript fully approved

Author Response

Reviewer 2 Report (Previous Reviewer 1)

Overall, the authors implemented the various comment that were raised in the previous submission.

The paper is interesting and relevant for the profession, especially with the growing needs and interests to reduce CO2 emissions. 

Nevertheless, the reviewer still believes that English editing and improvement are necessary to enhance the flow of ideas and better convey the message to readers in easier and more comprehensive way.

Author Response

Reviewer 3 Report (New Reviewer)

Dear Authors,

Your topic is interesting, but finally a reader can be confused. You presented 3 scenarios of water system renovations, while the title of your paper concerns whole buildings. You should put more effort on describing the presented scenarios, I miss information about costs. Please write more about  INCIS-Decima District, what it is, how many users, what types of buildings, what types of industry, services, etc. there are. Write more about electricity production there. Without it CO2 calculations are impossible.

Please define what is GHG before you start using an abbreviation (from line 16), same with RES (from line 132).

Please rewrite the sentence in lines 201-207. It is difficult to understand its meaning. Same with lines 209-214, 229-237, 243-251, 727-733.

Does the Table 1 contain the electricity consumption for water distribution? I miss such explanation.

Eq. 4 - write more about k7 calculations. Is this factor constant for the whole country or for every district? Write more about it.

Round 2

Reviewer 3 Report (New Reviewer)

Dear Authors,

I appreciate your revised version.

This manuscript is a resubmission of an earlier submission. The following is a list of the peer review reports and author responses from that submission.

Round 1

Reviewer 1 Report

The abstract is very broad, and should be better written to reflect the main objectives of the paper.

The introduction contains good information, however, needs to be better structured with enhanced flow of ideas and English writing. Most importantly, the paper objectives should be better highlighted to reflect the originality of the work and need to conduct it (this info normally should be written in the last section of the introduction). 

The title of Section 2 (Materials and methods) should be changed. 

Section 2.4.1 should be supported by relevant references, and more detailed.

In all discussion, the sentences are too long, making difficult the understanding of the idea that the authors wish to convey. A thorough English checking is thus required.

Section 2.4.3 should be much more concise.

Section 2.5: please elaborate more on how and the rational behind the creation of those 3 scenarios. Support discussion with references. 

Line 396, indeed, there is high variability of data, which puts in question the precision and accuracy of the reported results throughout the complete paper! 

Note that the energy efficiency in buildings can be optimized by different ways; yet, this has not been accounted for in the submitted paper.

Throughout the results section, the authors made many speculations which reduced the accuracy of discussion and statements mentioned. Table 4 and figure 4 should be better discussed, and the authors should clearly define their limitations including their dependency on the input data. 

The conclusion part is not well written; in fact, there are no clear/definite conclusions that can be drawn from this study (mainly speculations). 

n/a

Reviewer 2 Report

Interesting article, clear methodology.

I suggest just to have a more detailed conclusion explaining why the sutdy is relevant.

please try to have "bigger" graphs!
